# CAUSAL REASONING IN THE PRESENCE OF LATENT CONFOUNDERS VIA NEURAL ADMG LEARNING

**Matthew Ashman**[2*], **Chao Ma**[1*], **Agrin Hilmkil**[1], **Joel Jennings**[1], **Cheng Zhang**[1]
[1]Microsoft Research Cambridge     [2]University of Cambridge
mca39@cam.ac.uk ,{chaoma,agrinhilmkil,joeljennings,chezha}@microsoft.com

## ABSTRACT

Latent confounding has been a long-standing obstacle for causal reasoning from observational data. One popular approach is to model the data using acyclic directed mixed graphs (ADMGs), which describe ancestral relations between variables using directed and bidirected edges. However, existing methods using ADMGs are based on either linear functional assumptions or a discrete search that is complicated to use and lacks computational tractability for large datasets. In this work, we further extend the existing body of work and develop a novel gradient-based approach to learning an ADMG with non-linear functional relations from observational data. We first show that the presence of latent confounding is identifiable under the assumptions of bow-free ADMGs with non-linear additive noise models. With this insight, we propose a novel neural causal model based on autoregressive flows for ADMG learning. This not only enables us to determine complex causal structural relationships behind the data in the presence of latent confounding, but also estimate their functional relationships (hence treatment effects) simultaneously. We further validate our approach via experiments on both synthetic and real-world datasets, and demonstrate the competitive performance against relevant baselines.

## 1   INTRODUCTION

Learning causal relationships and estimating treatment effects from observational studies is a fundamental problem in causal machine learning, and has important applications in many areas of social and natural sciences (Pearl, 2010; Spirtes, 2010). They enable us to answer questions in causal nature; for example, *what is the effect on the expected lifespan of a patient if I increase the dose of X drug?* However, many existing methods of causal discovery and inference overwhelmingly rely on the assumption that all necessary information is available. This assumption is often untenable in practice. Indeed, an important, yet often overlooked, form of causal relationships is that of latent confounding; that is, when two variables have an unobserved common cause (Verma & Pearl, 1990). If not properly accounted for, the presence of latent confounding can lead to incorrect evaluation of causal quantities of interest (Pearl, 2009).

Traditional causal discovery methods that account for the presence of latent confoundings, such as the fast causal inference algorithm (FCI) (Spirtes et al., 2000) and its extensions (Colombo et al., 2012; Claassen et al., 2013; Chen et al., 2021), rely on uncovering an equivalence class of acyclic directed mixed graphs (ADMGs) that share the same conditional independencies. Without additional assumptions, however, these methods might return uninformative results as they cannot distinguish between members of the same Markov equivalence class (Bellot & van der Schaar, 2021). More recently, causal discovery methods based on structural causal models (SCMs) (Pearl, 1998) have been developed for latent confounding (Nowzohour et al., 2017; Wang & Drton, 2020; Maeda & Shimizu, 2020; 2021; Bhattacharya et al., 2021). By assuming that the causal effects follow specific functional forms, they have the advantage of being able to distinguish between members of the same Markov equivalence class (Glymour et al., 2019). Yet, existing approaches either rely on restrictive linear functional assumptions (Bhattacharya et al., 2021; Maeda & Shimizu, 2020; Bellot & van der Schaar, 2021), and/or discrete search over the discrete space of causal graphs (Maeda &

---

*Equal contribution. This work is done when Matthew Ashman was an intern at Microsoft Research.

Shimizu, 2021) that are computationally burdensome and unintuitive to use. As a result, modeling non-linear causal relationships between variables in the presence of latent confounders in a scalable way remains an outstanding task.

In this work, we seek to utilize recent advances in differentiable causal discovery (Zheng et al., 2018; Bhattacharya et al., 2021) and neural causal models (Lachapelle et al., 2019; Morales-Alvarez et al., 2022; Geffner et al., 2022) to overcome these limitations. Our core contribution is to extend the framework of differentiable ADMG discovery for linear models (Bhattacharya et al., 2021) to non-linear cases using neural causal models. This enables us to build scalable and flexible methods capable of discovering non-linear, potentially confounded relationships between variables and perform subsequent causal inference. Specifically, our contributions include:

1. **Sufficient conditions for ADMG identifiability with non-linear SCMs (Section 4).** We assume: i) the functional relationship follows non-linear additive noise SCM; ii) the effect of observed and latent variables do not modulate each other, and iii) all latent variables confound a pair of non-adjacent observed nodes. Under these assumptions, the underlying ground truth ADMG causal graph is identifiable. This serves as a foundation for designing ADMG identification algorithms for flexible, non-linear SCMs based on deep generative models.

2. **A novel gradient-based framework for learning ADMGs from observational data (Section 5).** Based on our theoretical results, we further propose Neural ADMG Learning (N-ADMG), a neural autoregressive-flow-based model capable of learning complex non-linear causal relationships with latent confounding. N-ADMG utilizes variational inference to approximate posteriors over causal graphs and latent variables, whilst simultaneously learning the model parameters via gradient-based optimization. This is more efficient and accurate than discrete search methods, allowing us to replace task-specific search procedures with general purpose optimizers.

3. **Empirical evaluation on synthetic and real-world datasets (Section 6).** We evaluate N-ADMG on a variety of synthetic and real-world datasets, comparing performance with a number of existing causal discovery and inference algorithms. We find that N-ADMG provides competitive or state-of-the-art results on a range of causal reasoning tasks.

## 2 RELATED WORK

**Causal discovery with latent confounding.** Constraint-based causal discovery methods in the presence of latent confounding have been well-studied (Spirtes et al., 2000; Zhang, 2008; Colombo et al., 2012; Claassen et al., 2013; Chen et al., 2021). Without further assumptions, these approaches can only identify a Markov equivalence class of causal structures (Spirtes et al., 2000). When certain assumptions are made on the data generating process in the form of SCMs (Pearl, 1998), additional constraints can help identify the true causal structure. In the most general case, additional non-parametric constraints have been identified (Verma & Pearl, 1990; Shpitser et al., 2014; Evans, 2016). Further refinement can be made through the assumption of stricter SCMs. For example, in the linear Gaussian additive noise model (ANM) case, Nowzohour et al. (2017) proposes a score-based approach for finding an equivalent class of bow-free acyclic path diagrams. Both Maeda & Shimizu (2020) and Wang & Drton (2020) develop Independence tests based approach for linear non-Gaussian ANM case, with Maeda & Shimizu (2021) extending this to more general cases.

**Differentiable characterization of causal discovery.** All aforementioned approaches employ a search over a discrete space of causal structures, which often requires task-specific search procedures, and imposes a computational burden for large-scale problems. More recently, (Zheng et al., 2018) proposed a differentiable constraint on directed acyclic graphs (DAG), and frames the graph structure learning problem as a differentiable constrained optimization task in the absence of latent confounders. This is further generalized to the latent confounding case (Bhattacharya et al., 2021) through differentiable algebraic constraints that characterize the space of ADMGs. Nonetheless, this work is limited in that it only considers linear Gaussian ANMs.

## 3 BACKGROUND

### 3.1 STRUCTURAL CAUSAL MODELS IN THE ABSENCE OF LATENT CONFOUNDINGS

Unlike constraint-based approaches such as the PC algorithm, SCMs capture the asymmetry between causal direction through functional assumptions on the data generating process, and have been central to many recent developments in causal discovery (Glymour et al., 2019). . Given a directed acyclic graph (DAG) $G$ on nodes $\{1, \ldots, D\}$, SCMs describe the random variable $\mathbf{x} = (x_1, \ldots, x_D)$ by $x_i = f_i(\mathbf{x}_{\mathrm{pa}(i;G)}, \epsilon_i)$, where $\epsilon_i$ is an exogenous noise variable that is independent of all other variables in the model, $\mathrm{pa}(i;G)$ denotes the set of parents of node $i$ in $G$, and $f_i$ describes how $x_i$ depends on its parents and the noise $\epsilon_i$. We focus on additive noise SCMs, commonly referred to as additive noise models (ANMs), which take the form

$$x_i = f_i(\mathbf{x}_{\mathrm{pa}(i;G)}, \epsilon_i) = f_i(\mathbf{x}_{\mathrm{pa}(i;G)}) + \epsilon_i, \quad \text{or} \quad \mathbf{x} = f_G(\mathbf{x}) + \boldsymbol{\epsilon} \quad \text{in vector form.} \tag{1}$$

This induces a joint observation distribution $p_\theta(\mathbf{x}^n | G)$, where $\theta$ denotes the parameters for functions $\{f_i\}$. Under the additive noise model in Equation 1, the DAG $G$ is identifiable assuming causal minimality and no latent confounders (Peters et al., 2014).

### 3.2 GRAPHICAL REPRESENTATION OF LATENT CONFOUNDERS USING ADMGS

One of the most widely-used graphical representations of causal relationships involving latent confounding is the so-called acyclic directed mixed graph (ADMGs). ADMGs are an extension of DAGs, that contain both directed edges ($\rightarrow$) and bidirected edges ($\leftrightarrow$) between variables. More concretely, the directed edge $x_i \rightarrow x_j$ indicates that $x_i$ is an ancestor of $x_j$, and the bidirected edge $x_i \leftrightarrow x_j$ indicates that $x_i$ and $x_j$ share a common, unobserved ancestor (Richardson & Spirtes, 2002; Tian & Pearl, 2002). An ADMG $G$ over a collection of $D$ variables $\mathbf{x} = (x_1, \ldots, x_D)$ can be described using two binary adjacency matrices: $G_D \in \mathbb{R}^{D \times D}$, for which an entry of 1 in position $(i, j)$ indicates the presence of the directed edge $x_i \rightarrow x_j$, and $G_B \in \mathbb{R}^{D \times D}$, for which an entry of 1 in position $(i, j)$ indicates the presence of the bidirected edge $x_i \leftrightarrow x_j$. Throughout this paper, we will use the graph notation $G$ to indicate the tuple $G = \{G_D, G_B\}$. When using ADMGs to represent latent confounding, causal discovery amounts to learning the matrices $G_D$ and $G_B$.

Similar to a DAG, an SCM can be specified to describe the causal relationships implied by an ADMG through the so-called *magnification* process. As formulated in (Peña, 2016), whenever a bidirected edge $x_i \leftrightarrow x_j$ is present according to $G_B$ in an ADMG, we will explicitly add a latent node $u_m$ to represent the latent parent (confounder) of $x_i \leftrightarrow x_j$. Then, the SCM of $x_i$ can be written as $\mathbf{x}$ by $x_i = f_i(\mathbf{x}_{\mathrm{pa}(i;G_D)}, \mathbf{u}_{\mathrm{pa}(i;G_B)}) + \epsilon_i$, where $\mathbf{u}_{\mathrm{pa}(i;G_B)}$ denotes the latent parents of $x_i$ in the set of all latent nodes $\mathbf{u} = (u_1, \ldots, u_M)$ added in the magnification process. In compact form, we can write:

$$[\mathbf{x}, \mathbf{u}] = f_G(\mathbf{x}, \mathbf{u}) + \boldsymbol{\epsilon}. \tag{2}$$

The 'magnified SCM' will serve as a practical device for learning ADMGs in this paper. Similar to DAGs, magnified SCMs induce an observational distribution on $\mathbf{x}$, denoted by $p_\theta(\mathbf{x}^n; G)$. Note that given an ADMG, the magnified SCM is not unique as latent variables may be shared. For example, $x_1 \leftrightarrow x_2, x_2 \leftrightarrow x_3, x_3 \leftrightarrow x_1$ can be magnified as both $\{x_1 \leftarrow u_1 \rightarrow x_2, x_2 \leftarrow u_2 \rightarrow x_3, x_3 \leftarrow u_3 \rightarrow x_1\}$ and $\{u_1 \rightarrow x_1, u_1 \rightarrow x_2, u_1 \rightarrow x_3\}$. Therefore, ADMG identifiability does not imply the structural identifiability of the magnified SCM. In this paper, we focus on ADMG identifiability.

### 3.3 DEALING WITH GRAPH UNCERTAINTY

Additive noise SCMs only guarantee DAG identifiability in the limit of infinite data. In the finite data regime, there is inherent uncertainty in the causal relationships. The Bayesian approach to causal discovery accounts for this uncertainty in a principled manner using probabilistic modeling (Heckerman et al., 2006), in which one's belief in the true causal graph is updated using Bayes' rule. This approach is grounded in causal decision theory, which states that rational decision-makers maximize the expected utility over a probability distribution representing their belief in causal graphs (Soto et al., 2019). We can model the causal graph $G$ jointly with observations $\mathbf{x}^1, \ldots, \mathbf{x}^N$ as

$$p_\theta(\mathbf{x}^1, \ldots, \mathbf{x}^N, G) = p(G) \prod_{n=1}^{N} p_\theta(\mathbf{x}^n | G) \tag{3}$$

where $\theta$ denotes the parameters of the likelihood relating the observations to the underlying causal graph. In general, the posterior distribution $p_\theta(G|\mathbf{x}^1, \ldots, \mathbf{x}^N)$ is intractable. One approach to circumvent this is variational inference (Jordan et al., 1999; Zhang et al., 2018), in which we seek an approximate posterior $q_\phi(G)$ that minimises the KL-divergence $\mathrm{KL}\left[q_\phi(G)||p_\theta(G|\mathbf{x}^1, \ldots, \mathbf{x}^N)\right]$. This can be achieved through maximization of the evidence lower bound (ELBO), given by

$$\mathcal{L}_{\mathrm{ELBO}} = \sum_{n=1}^{N} \mathbb{E}_{q_\phi(G)}\left[\log p_\theta(\mathbf{x}^n|G)\right] - \mathrm{KL}\left[q_\phi(G)||p(G)\right]. \tag{4}$$

An additional convenience of this approach is that the ELBO serves as a lower bound to the marginal log-likelihood $p_\theta(\mathbf{x}^1, \ldots, \mathbf{x}^N)$, so it can also be maximized with respect to the model parameters $\theta$ to find the parameters that approximately maximize the likelihood of the data (Geffner et al., 2022).

## 4 ESTABLISHING ADMGs IDENTIFIABILITY UNDER NON-LINEAR SCMs

To build flexible methods that are capable of discovering causal relationships under the presence of latent confounding, we first need to establish the identifiability of ADMGs under non-linear SCMs. The concept of structural identifiability of ADMGs is formalized in the following definition:

**Definition 1** (ADMG structural identifiability). *For a distribution $p_\theta(\mathbf{x}; G)$, the ADMG $G = \{G_D, G_B\}$ is said to be structurally identifiable from $p_\theta(\mathbf{x}; G)$ if there exists no other distribution $p_{\theta'}(\mathbf{x}; G')$ such that $G \neq G'$ and $p_\theta(\mathbf{x}; G) = p_{\theta'}(\mathbf{x}; G')$.*

Assuming that our model is correctly specified and $p_{\theta^0}(\mathbf{x}; G^0)$ denotes the true data generating distribution, then ADMG structural identifiability guarantees that if we find some $p_\theta(\mathbf{x}; G) = p_{\theta^0}(\mathbf{x}; G^0)$ (by e.g., maximum likelihood learning), we can recover $G = G^0$. In this section, we seek to establish sufficient conditions under which the ADMG identifiability is satisfied. Let $\mathbf{x} = (x_1, \ldots, x_D)$ be a collection of observed random variables, and $\mathbf{u} = (u_1, \ldots, u_M)$ be a collection of unobserved (latent) random variables. Our first assumption is that data generating process can be expressed as a specific non-linear additive noise SCM, in which the effect of the observed and latent variables do not modulate each other, see the functional form below:

**Assumption 1.** *We assume that the data generating process takes the form*

$$[\mathbf{x}, \mathbf{u}]^\top = f_{G_D, \mathbf{x}}(\mathbf{x}; \theta) + f_{G_B, \mathbf{u}}(\mathbf{u}; \theta) + \boldsymbol{\epsilon} \tag{5}$$

*where each element of $\boldsymbol{\epsilon}$ is independent of all other variables in the model and $\theta$ denotes the parameters of the non-linear functions $f_{G_D, \mathbf{x}}$ and $f_{G_B, \mathbf{u}}$.*

This assumption is one of the elements that separate us from previous work of (Bhattacharya et al., 2021; Maeda & Shimizu, 2020) (linearity is assumed) and (Maeda & Shimizu, 2021) (effects are assumed to be fully decoupled, $x_i = \sum_m f_{im}(x_m \in \mathbf{x}_{\mathrm{pa}(i;G_D)}) + \sum_k g_{ik}(u_k \in \mathbf{u}_{\mathrm{pa}(i;G_B)}) + \epsilon_i$). As discussed in Section 3.2, the discovery of an ADMG between observed variables amounts to the discovery of their ancestral relationships. Therefore, the mapping between ADMGs and their magnified SCMs is not one-to-one, which might cause issues when designing causal discovery methods using SCM-based approaches. To further simplify the underlying latent structure (without losing too much generality), our second assumption assumes that every latent variable is a parentless common cause of a pair of non-adjacent observed variables:

**Assumption 2** (Latent variables confound pairs of non-adjacent observed variables). *For each latent variable $u_k$ in the data generating process, there exists a non-adjacent pair $x_i$ and $x_j$ that is unique to $u_k$, such that $x_i \leftarrow u_k \rightarrow x_j$.*

Arguments for Assumption 2 are made by (Pearl & Verma, 1992), who show that this family of causal graphs is very flexible, and can produce the same conditional independencies amongst the observed variables in any given causal graph. Therefore, it has been argued that without loss of generality, we can assume latent variables to be exogenous, and have exactly two non-adjacent children. [1] This assumption also implies that we can specify a magnified SCM to the ADMG as shown in Section 3.2, which allows us to evaluate and optimize the induced likelihood later on.

---

[1]Meanwhile, Evans (2016) show that this graph family cannot induce all possible observational distributions. In Appendix E we show that in the linear non-Gaussian ANM case there exist certain constraints on marginal distributions that cannot be satisfied under Assumption 2. Nonetheless, we could not derive such constraints for the non-linear case.

Given Assumptions 1 and 2, we now provide three lemmas which allow us to identify the ADMG matrices, $G_D$ and $G_B$. These lemmas extend the previous results in Maeda & Shimizu (2021) under our new assumptions above. Note that all $g_i$ and $g_j$ functions in the lemmas denote functions satisfying the residual faithfulness condition (Maeda & Shimizu (2021)), as detailed in Appendix A.

**Lemma 1** (Case 1). *Given Assumptions 1 and 2, then $[G_B]_{i,j} = 1$ and $[G_D]_{i,j} = 0$ if and only if*

$$\forall g_i, g_j, \ [(x_i - g_i(\mathbf{x}_{-i}) \not\perp\!\!\!\perp (x_j - g_j(\mathbf{x}_{-j})]. \tag{6}$$

**Lemma 2** (Case 2). *Given Assumptions 1 and 2, then $[G_B]_{i,j} = 0$ and $[G_D]_{i,j} = 0$ if and only if*

$$\exists g_i, g_j, \ [(x_i - g_i(\mathbf{x}_{-(i,j)}) \perp\!\!\!\perp (x_j - g_j(\mathbf{x}_{-(i,j)}))]. \tag{7}$$

**Lemma 3** (Case 3). *Given Assumptions 1 and 2, then $[G_B]_{i,j} = 0$ and $[G_D]_{i,j} = 1$ if and only if*

$$\forall g_i, g_j, \ [(x_i - g_i(\mathbf{x}_{-(i,j)}) \not\perp\!\!\!\perp (x_j - g_j(\mathbf{x}_{-j})] \tag{8}$$

$$\exists g_i, g_j, \ [(x_i - g_i(\mathbf{x}_{-i}) \perp\!\!\!\perp (x_j - g_j(\mathbf{x}_{-(i,j)}))] \tag{9}$$

Since each case leads to mutually exclusive conditional independence/dependence constraints, for any $p_\theta(\mathbf{x}; G)$ specified under the assumptions above, there cannot exist some $p_{\theta'}(\mathbf{x}; G')$ such that $G \neq G'$ and $p_\theta(\mathbf{x}; G) = p_{\theta'}(\mathbf{x}; G')$, thus structural identifiability is satisfied. This gives our ADMG identifiability result:

**Proposition 1** (Identifiability of ADMGs under non-linear SCMs). *Assume Assumptions 1 to 2 hold. Then, $G_D$ and $G_B$ are identifiable for the data generating process specified in Equation 5.*

Whilst theoretically we could test for these conditions stated in Lemmas 1 to 3 directly, a more efficient approach is to use differentiable maximum likelihood learning. Assuming the model is correctly specified, then ADMG identifiability ensures the maximum likelihood estimate recovers the true graph in the limit of infinite data (Section 5.5). Hence, we can design the ADMG identification algorithms via maximum likelihood learning.

## 5 NEURAL ADMG LEARNING

Whilst we have outlined sufficient conditions under which the underlying ADMG causal structure can be identified, this does not directly provide a framework through which causal discovery and inference can be performed. In this section, we seek to formulate a practical framework for gradient-based ADMG identification. Three challenges that remain are: 1. How can we parameterize the magnified SCM models for ADMG to enable learning flexible causal relationships? 2. How can we optimise our model in the space of ADMG graphs as assumed in Section 4? 3. How do we learn the ADMG causal structure efficiently, whilst accounting for the missing data ($\mathbf{u}$) and graph uncertainties in the finite data regime? In this section, we present Neural ADMG Learning (N-ADMG), a novel framework that addresses all three challenges.

### 5.1 NEURAL AUTO-REGRESSIVE FLOW PARAMETERIZATION

We assume that our model used to learn ADMGs from data is correctly specified. That is, it can be written in the same magnified SCM form as in Equation 5:

$$[\mathbf{x}, \mathbf{u}]^\top = f_{G_D, \mathbf{x}}(\mathbf{x}; \theta) + f_{G_B, \mathbf{u}}(\mathbf{u}; \theta) + \boldsymbol{\epsilon}. \tag{10}$$

Following Khemakhem et al. (2020), we factorise the likelihood $p_\theta(\mathbf{x}^n, \mathbf{u}^n | G)$ induced by Equation 10 in an autoregressive manner. We can rearrange Equation 10 as

$$\boldsymbol{\epsilon} = \mathbf{v} - f_{G_D, \mathbf{x}}(\mathbf{x}; \theta) - f_{G_B, \mathbf{u}}(\mathbf{u}; \theta) := g_{\tilde{G}}(\mathbf{v}; \theta) = \mathbf{v} - f_{\tilde{G}}(\mathbf{v}; \theta) \tag{11}$$

where $\mathbf{v} = (\mathbf{x}, \mathbf{u}) \in \mathbb{R}^{D+M}$, and $\tilde{G}$ is the magnified adjacency matrix on $\mathbf{v}$, defined as $\tilde{G}_{i,j} \in \{0, 1\}$ if and only if $v_i \to v_j$. This allows us to express the likelihood as

$$p_\theta(\mathbf{v}^n | G) = p_{\boldsymbol{\epsilon}}(g_{\tilde{G}}(\mathbf{v}^n; \theta)) = \prod_{i=1}^{D+M} p_{\epsilon_i}(g_{\tilde{G}}(\mathbf{v}^n; \theta)_i). \tag{12}$$

Note that we have omitted the Jacobian-determinant term as it is equal to one since $\tilde{G}$ is acyclic (Mooij et al., 2011). Following Geffner et al. (2022), we adopt an efficient, flexible parameterization for the functions $f_i$ taking the form (consistent with Assumption 1)

$$f_i(\mathbf{v}) = \xi_{1,i}\left(\sum_{v_j \in \mathbf{x}}^{D} \tilde{G}_{j,i}\ell_j(v_j)\right) + \xi_{2,i}\left(\sum_{v_j \in \mathbf{u}}^{M} \tilde{G}_{j,i}\ell_j(v_j)\right) \tag{13}$$

where $\xi_{1,i}$, $\xi_{2,i}$ and $\ell_i$ ($i = 1, ..., D+M$) are MLPs. A naïve implementation would require training $3(D+M)$ neural networks. Instead, we construct these MLPs so that their weights are shared across nodes as $\xi_{1,i}(\cdot) = \xi_{1,i}(\mathbf{e}_i, \cdot)$, $\xi_{2,i}(\cdot) = \xi_{2,i}(\mathbf{e}_i, \cdot)$ and $\ell_i(\cdot) = \ell(\mathbf{e}_i, \cdot)$, with $\mathbf{e}_i \in \mathbb{R}^{D+M}$ a trainable embedding that identifies the output and input nodes respectively.

## 5.2 ADMG Learning via Maximizing Evidence Lower Bound

Our ADMG identifiability theory in Section 4 suggests that the ground truth ADMG graph can, in principle, be recovered via maximum likelihood learning of $p_\theta(\mathbf{x}|G)$. However, the aforementioned challenges remain: given a finite number of observations $\mathbf{x}^1, \ldots, \mathbf{x}^N$, how do we deal with the corresponding missing data $\mathbf{u}^1, \ldots, \mathbf{u}^N$ while learning the ADMG? How do we account for graph uncertainties and ambiguities in the finite data regime? To address these issues, N-ADMG takes a Bayesian approach toward ADMG learning. Similar to Section 3.3, we may jointly model the distribution over the ADMG causal graph $G$, the observations $\mathbf{x}^1, \ldots, \mathbf{x}^N$, and the corresponding latent variables $\mathbf{u}^1, \ldots, \mathbf{u}^N$, as

$$p_\theta(\mathbf{x}^1, \mathbf{u}^1, \ldots, \mathbf{x}^N, \mathbf{u}^N, G) = p(G)\prod_{n=1}^{N} p_\theta(\mathbf{x}^n, \mathbf{u}^n|G) \tag{14}$$

where $p_\theta(\mathbf{x}^n, \mathbf{u}^n|G)$ is the neural SCM model specified in Section 5.1, $\theta$ denotes the corresponding model parameters, and $p(G)$ is some prior distribution over the graph. Our goal is to learn both the model parameters $\theta$ and an approximation to the posterior $q_\phi(\mathbf{u}^1, \ldots, \mathbf{u}^N, G) \approx p_\theta(\mathbf{u}^1, \ldots, \mathbf{u}^N, G|\mathbf{x}^1, \ldots, \mathbf{x}^N)$. This can be achieved jointly using the variational inference framework (Zhang et al., 2018; Kingma & Welling, 2013), in which we maximize the evidence lower bound (ELBO) $\mathcal{L}_{\text{ELBO}}(\theta, \phi) \leq \sum_n \log p_\theta(\mathbf{x}^n)$ given by

$$\mathcal{L}_{\text{ELBO}}(\theta, \phi) = \mathbb{E}_{q_\phi(G)}\left[\sum_{n=1}^{N} \mathbb{E}_{q_\phi(\mathbf{u}^n|G)}\left[\log p_\theta(\mathbf{x}^n|\mathbf{u}^n, G)\right]\right] - \text{KL}\left[q_\phi(G)||p(G)\right]$$

$$- \mathbb{E}_{q_\phi(G)}\left[\sum_{n=1}^{N} \text{KL}\left[q_\phi(\mathbf{u}^n|\mathbf{x}^n, G))||p_\theta(\mathbf{u}^n|G)\right]\right]. \tag{15}$$

In the following sections, we describe our choice of the graph prior $p(G)$, and approximate posterior $q_\phi(\mathbf{u}^1, \ldots, \mathbf{u}^N, G) = \prod_n q_\phi(\mathbf{u}^n|\mathbf{x}^n, G)q_\phi(G)$. We will also demonstrate that maximizing $\mathcal{L}_{\text{ELBO}}(\theta, \phi)$ recovers the true ADMG causal graph in the limit of infinite data. See Section 5.5.

## 5.3 Choice of Prior Over ADMG Graphs

As discussed in Section 3.2, the ADMG $G$ can be parameterized by two binary adjacency matrices, $G_D$ whose entries indicate the presence of a directed edge, and $G_B$ whose edges indicate the presence of a bidirected edge. As discussed in Section 4, a necessary assumption for structural identifiability is that each latent variable is a parent-less confounder of a pair of non-adjacent observed variables. This further implies that the underlying ADMG must be bow-free (both a directed and a bidirected edge cannot exist between the same pair of observed variables). This constraint can be imposed by leveraging the bow-free constrain penalty introduced by Bhattacharya et al. (2021),

$$h(G_D, G_B) = \text{trace}\left(e^{G_D}\right) - D + \text{sum}\left(G_D \circ G_B\right) \tag{16}$$

which is non-negative and zero only if $(G_D, G_B)$ is a bow-free ADMG. As suggested in Geffner et al. (2022), we implement the prior as

$$p(G) \propto \exp\left(-\lambda_{s1}\|G_D\|_F^2 - \lambda_{s2}\|G_B\|_F^2 - \rho h(G_D, G_B)^2 - \alpha h(G_D, G_B)\right) \tag{17}$$

where the coefficients $\alpha$ and $\rho$ are increased whilst maximizing $\mathcal{L}_{\text{ELBO}}(\theta, \phi)$, following an augmented Lagrangian scheme (Nemirovski, 1999). Prior knowledge about the sparseness of the graph is introduced by penalizing the norms $\|G_D\|_F^2$ and $\|G_B\|_F^2$ with scaling coefficients $\lambda_{s1}$ and $\lambda_{s2}$.

## 5.4 Choice of Variational Approximation

We seek to approximate the intractable true posterior $p_\theta(\mathbf{u}^1, \ldots, \mathbf{u}^N, G | \mathbf{x}^1, \ldots, \mathbf{x}^N)$ using the variational distribution $q_\phi(\mathbf{u}^1, \ldots, \mathbf{u}^N, G)$. We assume the following factorized approximate posterior

$$q_\phi(\mathbf{u}^1, \ldots, \mathbf{u}^N, G) = q_\phi(G) \prod_{n=1}^{N} q_\phi(\mathbf{u}^n | \mathbf{x}^n). \tag{18}$$

For $q_\phi(G)$, we use a product of Bernoulli distributions for each potential directed edge in $G_D$ and bidirected edge in $G_B$. For $G_D$, edge existence and edge orientation are parameterized separately using the ENCO parameterization (Lippe et al., 2021). For $q_\phi(\mathbf{u}^n | \mathbf{x}^n)$, we apply amortized VI as in VAE literature (Kingma & Welling, 2013), where $q_\phi(\mathbf{u}^n | \mathbf{x}^n)$ is parameterized as a Gaussian distribution whose mean and variance is determined by passing the $\mathbf{x}^n$ through an inference MLP .

## 5.5 Maximizing $\mathcal{L}_{\text{ELBO}}(\theta, \phi)$ Recovers the Ground Truth ADMG

In Section 4, we have proved the structural identifiability of ADMGs. In this section, we further show that under certain assumptions, maximizing $\mathcal{L}_{\text{ELBO}}(\theta, \phi)$ recovers the true ADMG graph (denoted by $G^0$) in the infinite data limit. This result is stated in the following proposition:

**Proposition 2** (Maximizing $\mathcal{L}_{\text{ELBO}}(\theta, \phi)$ recovers the ground truth ADMG). *Assume that:*

- *Assumptions 1 and 2 (hence the identifiability of ADMGs) holds for the model $p_\theta(\mathbf{x}; G)$.*

- *The model is correctly specified ($\exists \theta^*$ such that $p_{\theta^*}(\mathbf{x}; G^0)$ recovers the data-generating process).*

- *Regularity condition: for all $\theta$ and $G$ we have $\mathbb{E}_{p(\mathbf{x}; G^0)}\left[|\log p_\theta(\mathbf{x}; G)|\right] < \infty$.*

- *The variational family of $q_\phi(\mathbf{u} | \mathbf{x}, G)$ is flexible enough, i.e., it contains $p_\theta(\mathbf{u} | \mathbf{x}, G)$.*

*Then, the solution $(\theta', q'_\phi(G))$ that maximizes $\mathcal{L}_{ELBO}(\theta, \phi)$ satisfies $q'_\phi(G) = \delta(G = G^0)$.*

The proof of Proposition 2 can be found in Appendix B, which justifies performing causal discovery by maximizing ELBO of the N-ADMG model. Once the model has been trained and the ADMG has been recovered, we can use the N-ADMG to perform causal inference as detailed in Appendix C.

## 6 Experiments

We evaluate N-ADMG in performing both causal discovery and causal inference on a number of synthetic and real-world datasets. Note that we run our model *both with and without the bow-free constraint, identified as N-BF-ADMG (our full model) and N-ADMG (for ablation purpose)*, respectively. We compare the performance of our model against five baselines: DECI (Geffner et al., 2022) (which we refer to as N-DAG for consistency), FCI (Spirtes et al., 2000), RCD (Maeda & Shimizu, 2020), CAM-UV (Maeda & Shimizu, 2021), and DCD (Bhattacharya et al., 2021). We evaluate the causal discovery performance using F1 scores for directed and and bidirected adjacency. The expected values of these metrics are reported using the learned graph posterior (which is deterministic for RCD, CAM-UV, and DCD). Causal inference is evaluated using the expected ATE as described in Appendix C. We evaluate the causal inference performance of the causal discovery benchmarks by fixing $q(G)$ to either deterministic or uniform categorical distributions on the learned causal graphs, then learning a non-linear flow-based ANM by optimizing Equation 15 in an identical manner to N-ADMG. A full list of results and details of the experimental set-up are included in Appendix F. Our implementation will be available at `https://github.com/microsoft/causica`.

### 6.1 Synthetic Fork-Collider Dataset

We construct a synthetic fork-collider dataset consisting of five nodes (Figure 1a). The data-generating process is a non-linear ANM with Gaussian noise. Variable pairs $(x_2, x_3)$, and $(x_3, x_4)$ are latent-confounded, whereas $(x_4, x_5)$ share a observed confounder. We evaluate both causal discovery and inference performances. For discovery, we evaluate F-score measure on both $G_D$ and $G_B$. For causal inference, we choose $x_4$ as the treatment variable taking and $x_2$, $x_3$ and $x_5$ as the

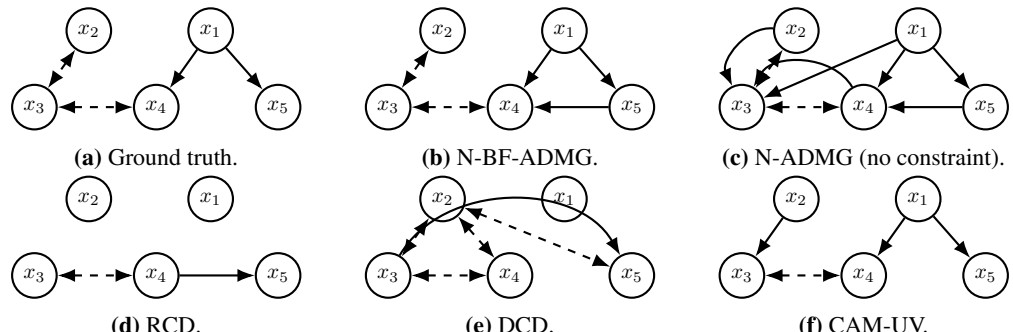

**Figure 1:** ADMG identification results on fork-collider dataset.

response variables, and evaluate ATE RMSE to benchmark performance. It is therefore crucial that discovery methods do not misidentify latent confounded variables as having a direct cause between them, as this would result in biased ATE estimates.

Figure 1b-1f visualizes the causal discovery results of different methods, and Table 1 summarizes both discovery and inference metrics. While no methods can fully recover the ground truth graph, our method N-BF-ADMG-G and CAM-UV achieved the best overall performance. N-BF-ADMG-G and N-ADMG-G on average are able to recover all bidirected edges from the data, while CAM-UV can only recover half of the latent variables. Without the bow-free constraint, N-ADMG-G discovers a di-

**Table 1:** Causal discovery and inference results for the fork-collider dataset. The table shows the mean and standard error results across five different random seeds.

| METHOD | $G_D$ FSCORE | $G_B$ FSCORE | ATE RMSE |
|---|---|---|---|
| N-BF-ADMG-G | 0.64 (0.06) | **0.93 (0.07)** | **0.022 (0.003)** |
| N-ADMG-G | 0.49 (0.02) | **0.99 (0.00)** | 0.239 (0.067) |
| N-DAG-G | 0.50 (0.00) | 0.00 (0.00) | **0.046 (0.025)** |
| FCI | 0.00 (0.00) | 0.75 (0.00) | 0.072 (0.015) |
| RCD | 0.00 (0.00) | 0.54 (0.00) | 0.206 (0.029) |
| CAM-UV | **0.80 (0.00)** | 0.67 (0.00) | **0.017 (0.003)** |
| DCD | 0.00 (0.00) | 0.67 (0.00) | 0.208 (0.064) |

rected edge from $x_4$ to $x_3$, which results in poor ATE RMSE performance. On the other hand, DAG-based method (N-DAG-G) is not able to deal with latent confounders, resulting in the poor f-scores in both $G_D$ and $G_B$. Linear ANM-based methods (RCD and DCD) perform significantly worse than other methods, resulting in 0 f-scores for directed matrices and largest ATE errors. This demonstrates the necessity for introducing non-linear assumptions.

### 6.2 RANDOM CONFOUNDED ER SYNTHETIC DATASET

We generate synthetic datasets from ADMG extension of *Erdős-Rényi* (ER) graph model (Lachapelle et al., 2019). We first sample random ADMGs from ER model, and simulate each variable using a randomly sampled nonlinear ANM. Latent confounders are then removed from the training set. See Appendix G.2 for details. We consider the number of nodes, directed edges and latent confounders triplets $(d, e, m) \in \{(4, 6, 2), (8, 20, 6), (12, 50, 10)\}$. The resulting datasets are identified as **ER**$(d, e, m)$. Figure 2 compares the performance of N-ADMG with the baselines. All variants of N-ADMG outperform the baselines for most datasets, highlighting its effectiveness relative to other methods (even those that employ similar assumptions). Similar to the fork-collider dataset, we see that methods operating under the assumption of linearity perform poorly when the data-generating process is non-linear. It is worth noting that even when the exogenous noise of N-ADMG is misspecified, its still exceeds that of other methods in most cases. This robustness is a desirable property as in many settings the form of exogenous noise is unknown.

### 6.3 INFANT HEALTH AND DEVELOPMENT PROGRAM (IHDP) DATASET

For the real-world datasets, we evaluate treatment effect estimation performances on infant health and development program data (IHDP). This dataset contains measurements of both infants and their mother during real-life data collected in a randomized experiment. The main task is to estimate the effect of home visits by specialists on future cognitive test scores of infants, where the ground

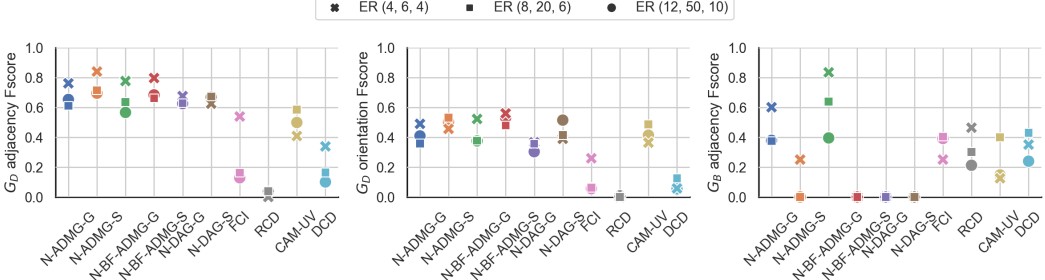

**Figure 2:** Causal discovery results for synthetic ER datasets. For readability, the N-ADMG results are connected with lines. The figure shows mean results across five randomly generated datasets.

truth outcomes are simulated as in (Hill, 2011). To make the task more challenging, additional confoundings are introduced by removing a subset (non-white mothers) of the treated population. More details can be found in Appendix G.3. We first perform causal discovery to learn the underlying ADMG of the dataset, and then perform causal inference. Since the true causal graph is unknown, we evaluate the causal inference performance of each method by estimating the ATE RMSE.

Apart from the aforementioned baselines, here we introduce four more methods: PC-DWL (PC algorithm for discovery, DoWhy (Sharma et al., 2021) linear adjustment for inference); PC-DwNL (PC for discovery, DoWhy double machine learning for inference); N-DAG-G-DwL (N-DAG Gaussian for discovery, linear adjustment for inference); and N-DAG-S-DwL (N-DAG Spline for discovery, linear adjustment for inference). Results are summarized in Figures 3a to 3c. Generally, models with non-Gaussian exogenous assumptions tend to have lower ATE estimation errors; while models with linear assumptions (RCD and DCD) have the worst ATE RMSE. Interestingly, the DoWhy-based plug-in estimators tend to worsen the performances of SCM models. However, regardless of the assumptions made on exogenous noise, our method (N-BF-ADMG-G and N-BF-ADMG-S) consistently outperforms all other baselines with the same noise. It is evident that for causal inference in real-world datasets, the ability of N-BF-ADMG to handle latent confoundings and nonlinear causal relationships becomes very effective.

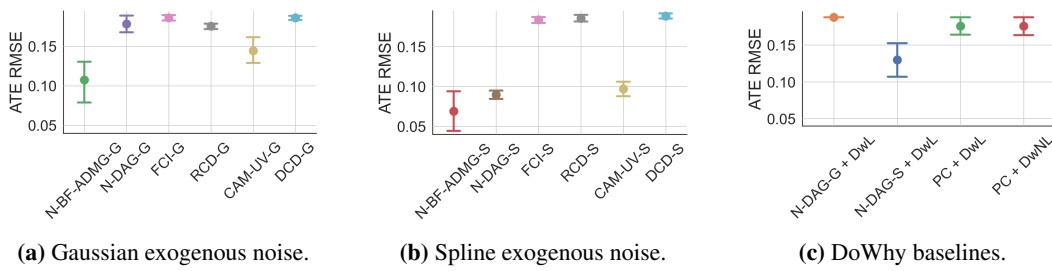

**(a)** Gaussian exogenous noise.     **(b)** Spline exogenous noise.     **(c)** DoWhy baselines.

**Figure 3:** Causal inference results for the IHDP dataset. The figure shows mean ± standard error results across five random initialisations.

## 7 Conclusion And Future Work

In this work, we proposed Neural ADMG Learning (N-ADMG), a novel framework for gradient-based causal reasoning in the presence of latent confounding for nonlinear SCMs. We established identifiability theory for nonlinear ADMGs under latent confounding, and proposed a practical ADMG learning algorithm that is both flexible and efficient. In future work, we will further extend our framework on how to more general settings (e.g., the effect from observed and latent variables can modulate in certain forms; latent variables can confound adjacent variables; etc), and how to improve certain modelling choices such as variational approximation qualities over both causal graph and latent variables

## Reproducibility Statement

A number of efforts have been made/will be made for the sake of reproducibility. First, we open source our package in our github page `github.com/microsoft/causica/tree/v0.0.0`. In addition, in this paper we included clear explanations of any assumptions and a complete proof of the claims in the Appendix, as well as model settings and hyperparameters. All datasets used in this paper are either publicly available data, or synthetic data whose generation process is described in detail.

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

# APPENDIX

## A  PROOF OF LEMMAS 1 TO 3

First we describe the residual faithfulness condition inherited from (Maeda & Shimizu, 2021) in our Lemmas 1 to 3:

**Definition 2** (Residual faithfulness condition). *We say that nonlinear functions $g_i, g_j$ satisfies the residual faithfulness condition, if: for any two arbitrary subset of $\mathbf{x}$, denote by $M$ and $N$, when both $(x_i - g_i(M))$ and $(x_i - g_j(M))$ have terms involving the same exogenous noise $\epsilon_k$, then $(x_i - g_i(M))$ and $(x_i - g_j(M))$ are mutually dependent.*

Next, we provide the proof for Lemmas 1 to 3 in Section 4. To prove those lemmas, we need the help of the following lemma, which extends Lemma A in (Maeda & Shimizu, 2021), except we don't assume any form for $g_i$ other than non-linearity.

**Lemma 4.** *Let $s(x_i)$ denote an arbitrary function of $x_i$. The residual of $s(x_i)$ regressed onto $\mathbf{x}_{-i}$ cannot be independent of $\epsilon_i$:*

$$\forall g_i, \ [s(x_i) - g_i(\mathbf{x}_{-i}) \not\perp\!\!\!\perp \epsilon_i]. \tag{19}$$

*Proof.* Assume that $[s(x_i) - g_i(\mathbf{x}_{-i}) \perp\!\!\!\perp \epsilon_i]$ holds, then $\mathbf{x}_{-i}$ must contain at least one descendent of $x_i$ as it must have dependence on the noise $\epsilon_i$ to cancel effect of $\epsilon_i$ in $s(x_i)$. We can express $g_i(\mathbf{x}_{-i})$ as $u_i(\boldsymbol{\epsilon})$. Since $g_i$ operates on variables defined by non-linear transformations of the exogenous noise terms, we cannot express $u_i$ as $a_i(\boldsymbol{\epsilon}_{-i}) + b_i(\epsilon_i)$. $\mathbf{x}_{-i}$ contains a descendent of $x_i$, so $\boldsymbol{\epsilon}_{-i}$ includes at least one noise term $\epsilon_k$ that satisfies $x_i \perp\!\!\!\perp \epsilon_k$ (i.e. is not in $x_i$). Thus, terms containing $\epsilon_i$ cannot be fully removed from $s(x_i) - g_i(\mathbf{x}_{-i})$ and so $[s(x_i) - g_i(\mathbf{x}_{-i}) \perp\!\!\!\perp \epsilon_i]$ does not hold. $\square$

### A.1  PROOF OF LEMMA 1

*Proof.* Define $g_i$ and $g_j$ as $g_i(\mathbf{x}_{-i}) = f_{i,\mathbf{x}}(\mathbf{pa}_{\mathbf{x}}(i)) + g_i'(\mathbf{x}_{-i})$ and $g_j(\mathbf{x}_{-j}) = f_{j,\mathbf{x}}(\mathbf{pa}_{\mathbf{x}}(j)) + g_i'(\mathbf{x}_{-j})$ respectively. Then, Equation 6 becomes equivalent to

$$\forall g_i', g_j', \ [(f_{i,\mathbf{u}}(\mathbf{pa}_{\mathbf{u}}(i)) + \epsilon_i - g_i'(\mathbf{x}_{-i}) \not\perp\!\!\!\perp (f_{j,\mathbf{u}}(\mathbf{pa}_{\mathbf{u}}(j)) + \epsilon_j - g_j'(\mathbf{x}_{-j})]. \tag{20}$$

Given Lemma 4 and following the same arguments as in Maeda & Shimizu (2021), this is equivalent to

$$(f_{i,\mathbf{u}}(\mathbf{pa}_{\mathbf{u}}(i)) + \epsilon_i) \not\perp\!\!\!\perp (f_{j,\mathbf{u}}(\mathbf{pa}_{\mathbf{u}}(j)) + \epsilon_j). \tag{21}$$

Since $\epsilon_i \perp\!\!\!\perp \epsilon_j$, we have

$$(f_{i,\mathbf{u}}(\mathbf{pa}_{\mathbf{u}}(i)) \not\perp\!\!\!\perp \epsilon_j) \vee (n_i \not\perp\!\!\!\perp f_{j,\mathbf{u}}(\mathbf{pa}_{\mathbf{u}}(j))) \vee (f_{i,\mathbf{u}}(\mathbf{pa}_{\mathbf{u}}(i)) \not\perp\!\!\!\perp f_{j,\mathbf{u}}(\mathbf{pa}_{\mathbf{u}}(j))). \tag{22}$$

The first implies the existence of an unobserved mediator between $x_j$ and $x_i$, the second implies the existence of an unobserved mediator between $x_i$ and $x_j$, and the third implies the existence of an unobserved confounder. Given the assumption of latent variables being confounders and minimality, this indicates the presence of a latent confounder and no direct cause between $x_i$ and $x_j$. $\square$

### A.2  PROOF OF LEMMA 2

*Proof.* When Equation 7 holds, Equation 6 does not. Thus, there is no unobserved confounder between $x_i$ and $x_j$. Assume that $x_j$ is a direct cause of $x_i$, and that Equation 7 is satisfied for $g_i$ and $g_j$. $x_i$ contains a nonlinear function of $\epsilon_j$ that cannot be removed by $g_i(\mathbf{x}_{-(i,j)})$, thus $(x_i - g_i(\mathbf{x}_{-(i,j)})) \not\perp\!\!\!\perp \epsilon_j$. Similarly, $(x_j - g_j(\mathbf{x}_{-(i,j)})) \not\perp\!\!\!\perp \epsilon_i$. Thus, we have $[(x_i - g_i(\mathbf{x}_{-(i,j)})) \not\perp\!\!\!\perp (x_j - g_j(\mathbf{x}_{-(i,j)}))]$ which contradicts our initial assumption. The same arguments apply when $x_i$ is a direct cause of $x_j$, implying that there can be no causal relationship between $x_i$ and $x_j$. $\square$

### A.3  PROOF OF LEMMA 3

*Proof.* When Equation 9 holds, Equation 6 does not and so there is no latent confounder between $x_i$ and $x_j$. When *Equation* 8 holds, Equation 7 does not hold and so there is a direct causal relationship between $x_i$ and $x_j$.

Assume that $x_j$ is a direct cause of $x_i$. Define $g_1(\mathbf{x}_{-i}) = f_{i,\mathbf{x}}(\mathbf{pa_x}(i))$ and $g_2(\mathbf{x}_{-(i,j)}) = f_{j,\mathbf{x}}(\mathbf{pa_x}(j))$, giving $x_i - g_i(\mathbf{x}_{-i}) = f_{i,\mathbf{u}}(\mathbf{pa_u}(i)) + \epsilon_i$ and $x_j - g_j(\mathbf{x}_{-(i,j)}) = f_{j,\mathbf{u}}(\mathbf{pa_u}(j)) + \epsilon_j$. When there is no latent confounder, $f_{i,\mathbf{u}}(\mathbf{pa_u}(i)) \perp\!\!\!\perp f_{j,\mathbf{u}}(\mathbf{pa_u}(j))$. Thus, $(f_{i,\mathbf{u}}(\mathbf{pa_u}(i)) + \epsilon_i) \perp\!\!\!\perp (f_{j,\mathbf{u}}(\mathbf{pa_u}(j)) + \epsilon_j)$ hods and Equation 9 is satisfied.

Now, assume that $x_i$ is a direct cause of $x_j$. Using Lemma 4, we have

$$\forall g_i, \; [(x_i - g_i(\mathbf{x}_{-i}) \not\perp\!\!\!\perp \epsilon_i]. \tag{23}$$

Similarly, since $x_j$ is a function of $x_i$ we also have

$$\forall g_j, \; [(x_j - g_j(\mathbf{x}_{-(i,j)}) \not\perp\!\!\!\perp \epsilon_i]. \tag{24}$$

Collectively, this implies

$$\forall g_i, g_j, \; [(x_i - g_i(\mathbf{x}_{-i}) \not\perp\!\!\!\perp (x_j - g_j(\mathbf{x}_{-i,j})] \tag{25}$$

which contradicts Equation 9. Thus, if Equation 9 is satisfied then $x_j$ is a direct cause of $x_i$. $\qquad\square$

## B  PROOF OF PROPOSITION 2

To prove Proposition 2, we need the following lemma:

**Lemma 5.** *Assume a variational distribution $q_\phi(G)$ over a space of graphs $\mathcal{G}_\phi$, where each graph $G \in \mathcal{G}_\phi$ has a non-zero associated weight $w_\phi(G)$. With the soft prior $p(G)$ defined as Equation 17 and bounded $\lambda_1, \lambda_2, \rho, \alpha$, we have*

$$\lim_{N \to \infty} \frac{1}{N} \mathrm{KL}[q_\phi(G)\|p(G)] = 0. \tag{26}$$

*Proof.* This directly follows from Lemma 1 of (Geffner et al., 2022). $\qquad\square$

Now we can proceed to prove Proposition 2:

*Proof.* For N-ADMG, in the infinite data limit $\mathcal{L}_{\mathrm{ELBO}}$ becomes

$$\lim_{N \to \infty} \frac{1}{N} \sum_{n=1}^{N} \mathbb{E}_{q_\phi(G)q(\mathbf{u}_n|G)} \left[\log p_\theta(\mathbf{x}_n, \mathbf{u}_n|G)\right] - \frac{1}{N} \sum_{n=1}^{N} H[q(\mathbf{u}_n|\mathbf{x}_n, G)] - \underbrace{\frac{1}{N}\mathrm{KL}\left[q(G)\|p(G)\right]}_{\to 0}$$

$$= \lim_{N \to \infty} \frac{1}{N} \sum_{n=1}^{N} \sum_{G \in \mathcal{G}_\phi} w_\phi(G) \mathbb{E}_{q(\mathbf{u}|\mathbf{x},G)} \left[\log p_\theta(\mathbf{x}_n|\mathbf{u}_n, G)\right] - \frac{1}{N} \sum_{n=1}^{N} \mathbb{E}_{q(G)} \left[\mathrm{KL}\left[q(\mathbf{u}_n|\mathbf{x}_n G)\|p(\mathbf{u}_n|\mathbf{x}_n, G)\right]\right]. \tag{27}$$

where the zeroing of the KL divergence follows from Lemma 5. Given fixed $\theta$, the optimal posterior $q^*(\mathbf{u}_n|\mathbf{x}_n, G)$ satisfies $q^*(\mathbf{u}_n|\mathbf{x}_n, G) = p_\theta(\mathbf{u}_n|\mathbf{x}_n, G)$ due to the flexibility assumption. Thus,

$$\lim_{N \to \infty} \mathcal{L}_{\mathrm{ELBO}}(\theta, \phi, q^*(\mathbf{u}_n|\mathbf{x}_n, G))$$

$$= \lim_{N \to \infty} \frac{1}{N} \sum_{n=1}^{N} \sum_{G \in \mathcal{G}_\phi} w_\phi(G) \log p_\theta(\mathbf{x}_n|G) \tag{28}$$

$$= \int p(\mathbf{x}; G^0) \sum_{G \in \mathcal{G}_\phi} w_\phi(G) \log p_\theta(\mathbf{x}|G) d\mathbf{x}$$

where $p(\mathbf{x}; G^0)$ denotes the data generation distribution with the ground truth ADMG, $G^0$. Let $(\theta^*, G^*) = \arg\max \int p(\mathbf{x}; G^0) \log p_\theta(\mathbf{x}|G) d\mathbf{x}$ be the MLE solution. Since $\sum_{G \in \mathcal{G}_\phi} w_\phi(G) = 1$, $w_\phi(G) > 0$, we have

$$\sum_{G \in \mathcal{G}_\phi} w_\phi(G) \mathbb{E}_{p(\mathbf{x}; G^0)} \left[\log p_\theta(\mathbf{x}|G)\right] \leq \mathbb{E}_{p(\mathbf{x}; G^0)} \left[\log p_{\theta^*}(\mathbf{x}; G^*)\right]$$

with the optimal value of $\sum_{G \in \mathcal{G}_\phi} w_\phi(G) \mathbb{E}_{p(\mathbf{x};G^0)} [\log p_\theta(\mathbf{x}|G)]$ is achieved when every graph $G \in \mathcal{G}_\phi$ and associated parameter $\theta_G$ satisfies

$$\mathbb{E}_{p(\mathbf{x};G^0)} [\log p_{\theta_G}(\mathbf{x}|G)] = \mathbb{E}_{p(\mathbf{x};G^0)} [\log p_{\theta^*}(\mathbf{x}|G^*)] . \tag{29}$$

Since the model is correctly specified, the MLE solution $(\theta^*, G^*)$ satisfies

$$\mathbb{E}_{p(\mathbf{x};G^0)} [\log p_{\theta^*}(\mathbf{x}|G^*)] = \mathbb{E}_{p(\mathbf{x};G^0)} [\log p(\mathbf{x};G^0)]$$

Therefore, condition Equation 29 implies for every graph $G' \in \mathcal{G}_\phi$, $G' = G^0$ under the regularity condition; or equivalently, $\mathcal{G}_\phi = \{G' = G^0\}$. This proves our statement that $q'_\phi(G) = \delta(G = G')$, where $G' = G^0$.

$\square$

## C    ESTIMATING TREATMENT EFFECTS

For all experiments we consider, the causal quantity of interest we wish to estimate is the expected average treatment effect (ATE), $\mathbb{E}_{q_\phi(G)} [\text{ATE}(\mathbf{a}, \mathbf{b}|G)]$, where the expectation is taken with respect to our learned posterior over causal graphs $q_\phi(G)$:

$$\mathbb{E}_{q_\phi(G)} [\text{ATE}(\mathbf{a}, \mathbf{b}|G)] = \mathbb{E}_{q_\phi(G)} \left[ \mathbb{E}_{p(\mathbf{x}_Y|\text{do}(\mathbf{x}_T=\mathbf{b}),G)} [\mathbf{x}_Y] - \mathbb{E}_{p(\mathbf{x}_Y|\text{do}(\mathbf{x}_T=\mathbf{b}),G)} [\mathbf{x}_Y] \right] . \tag{30}$$

This requires samples from $p(\mathbf{x}_Y|\text{do}(\mathbf{x}_T = \mathbf{b}), G) = p(\mathbf{x}_Y|\mathbf{x}_T = \mathbf{b}, G_{\text{do}(\mathbf{x}_T)})$, where $G_{\text{do}(\mathbf{x}_T)}$ is the 'mutilated' graph obtained by removing incoming edges into $\mathbf{x}_T$. We can achieve this by simulating the learnt SCM on $G_{\text{do}(\mathbf{x}_T)}$ whilst keeping $\mathbf{x}_T = \mathbf{b}$ fixed. Note that $q_\phi(\mathbf{u}|\mathbf{x})$ is not used to estimate the ATE; it suffices to sample $\mathbf{u}$ from the prior distribution, $p(\mathbf{u})$. In our setting, the inference MLP $q_\phi(\mathbf{u}|\mathbf{x})$ is only used as a means through which the likelihood of the data can be evaluated efficiently, and thus model parameters learned. This is in a similar spirit to VAEs (Kingma & Welling, 2013).

## D    RELATED WORK ON CAUSAL INFERENCE UNDER LATENT CONFOUNDING

Attempting to perform causal inference in the presence of latent confounding can lead to biased estimates (Pearl, 2012). Whilst the observed data distribution may still be identifiable, estimating causal effects are not (Spirtes et al., 2000). A recent string of work has made progress in the case where the effects of multiple interventions are being estimated (Tran & Blei, 2017; Ranganath & Perotte, 2018; Wang & Blei, 2019; D'Amour, 2019). An alternative approach is to assume identifiability of the joint distribution over both latent and observed variables given just the observations. Louizos et al. (2017) point out that there are many cases in which this is possible (Khemakhem et al., 2020; Kingma & Welling, 2013). Nevertheless, all these methods assume the underlying causal graph is known. More recently, Mohammad-Taheri et al. (2021) argue that a DAG latent variable model trained on data can be used for down-stream causal inference tasks even if its parameters are non-identifiable, as long as the query can be identified from the observed variables according to the do-calculus.

## E    ADDITIONAL DISCUSSIONS ON STRUCTURAL IDENTIFIABILITY OF LATENT VARIABLES

In Section 3.2, we argued that the identifiability of ADMG does not imply the structural identifiability of the magnified SCM. In this section, we will present more discussions on certain identifiability of latent structures (of magnified SCMs). In general, these examples demonstrate that for linear non-Gaussian ANMs the structure of latent variables can be refined beyond the assumption of latent confounders acting between pairs of non-adjacent observed variables, whilst the same techniques cannot achieve the same for non-linear ANMs.

Recently, Cai et al. (2019) demonstrated that it is possible to discover the structure amongst latent variables using their so-called Triad constraints. Their method is limited to the linear non-Gaussian ANM case. In this section, we demonstrate that an analogous constraint isn't available for non-linear ANMs.

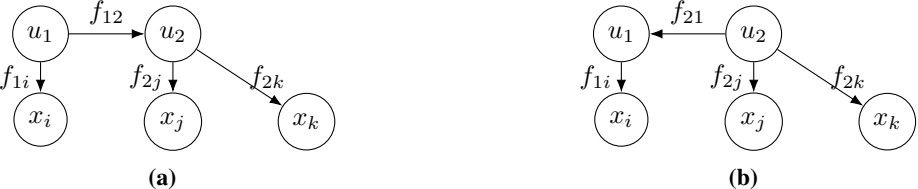

**Figure 4:** The two possible latent variable structures considered by Cai et al. (2019).

Consider the causal graphs shown in Figure 4a and Figure 4b.

**Lemma 6** (Linear non-Gaussian identifiability). *In the linear non-Gaussian ANM case, Equation 31 is satisfied only for the causal graph shown in Figure 4b. In the non-linear ANM case, Equation 31 is not satisfied for either Figure 4a or Figure 4b.*

$$\exists g, \quad x_i - g(x_j) \perp\!\!\!\perp x_k \tag{31}$$

*Sketch of Proof.* For the causal graph in Figure 4a, Equation 31 is equivalent to

$$\exists g \quad f_{1i}(u_1) + \epsilon_i - g(f_{2j}(f_{12}(u_1), u_2), \epsilon_j) \perp\!\!\!\perp f_{2k}(f_{12}(u_1), u_2) + \epsilon_k. \tag{32}$$

On the right, we have some function of the latent variables $u_1$ and $u_2$, $f_{2k}(f_{12}(u_1), u_2)$. On the left, we have some function of the same noise terms, $f_{1i}(u_1) - g(f_{2j}(f_{12}(u_1), \epsilon_j)$. To remove $u_1$ from both sides, we require $g$ to be non-zero and so a term including $u_2$ is still present. To remove $u_2$, we again require $g$ to be non-zero and so a term including $u_1$ is still present. Thus, Equation 31 is not satisfied in either the linear non-Gaussian or non-linear ANM case.

For the causal graph in Figure 4b, Equation 31 is equivalent to

$$\exists g \quad f_{1i}(f_{21}(u_2), u_1) + \epsilon_i - g(f_{2j}(u_2) + \epsilon_j) \perp\!\!\!\perp f_{2k}(u_2) + \epsilon_k. \tag{33}$$

In the linear case, we can construct a linear $g$ to remove $u_2$ fom the left side so that Equation 31 holds (i.e. $g = \frac{f_{1i}f_{21}}{f_{2j}}$). In the non-linear case, $u_1$ and $u_2$ are coupled in the leftmost term and cannot be removed by a term involving $u_2$ and $\epsilon_j$. Hence, Equation 31 is violated. $\qquad\square$

### E.2 Determining the Number of Latent Confounders

Here, we consider whether the number of latent confounders can be determined in the non-linear ANM case. Lemma 7 shows that in the linear non-Gaussian case, the number of latent confounders acting between a triplet of confounded observed variables can be determined (by verifying certain constraints on marginal distributions on observed variables), whilst the same approach cannot be used in the non-linear ANM case.

Consider the two causal graphs shown in Figure 5a and Figure 5b.

**Lemma 7** (Linear non-Gaussian identifiability). *In the linear non-Gaussian ANM case, Equation 34 is satisfied only for the causal graph shown in Figure 4b. In the non-linear ANM case, Equation 34 is not satisfied for either Figure 4a or Figure 4b.*

$$\exists g, \quad x_i - g(x_j) \perp\!\!\!\perp x_k. \tag{34}$$

*Sketch of Proof.* For Figure 5b, Equation 34 is equivalent to

$$\exists g \quad f_i(u) + \epsilon_i - g(f_j(u) + \epsilon_j) \perp\!\!\!\perp f_k(u) + \epsilon_k. \tag{35}$$

In the linear non-Gaussian case, it is straightforward to set $g = \frac{f_i}{f_j}$ to remove the common noise term $u$ from the left term and make the two sides independent. In the non-linear case, when $f_i \neq f_j$ the common noise term $u$ cannot be removed from the left as $g$ must be non-linear, and thus produces a term that involves both $u$ and $\epsilon_j$. For Figure 5a, Equation 34 is equivalent to

$$\exists g \quad f_i(u_1, u_2) + \epsilon_i - g(f_j(u_1, u_3) + \epsilon_j) \perp\!\!\!\perp f_k(u_2, u_3) + \epsilon_k. \tag{36}$$

$u_2$ cannot be removed from the left, and so Equation 34 does not hold in either the linear non-Gaussian or non-linear case. $\qquad\square$

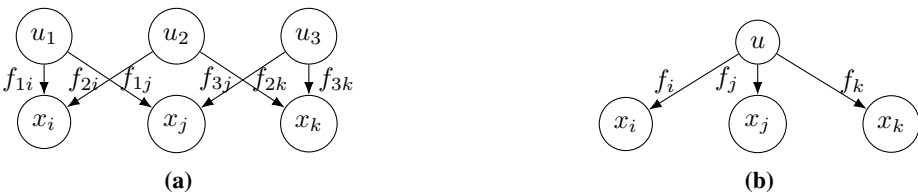

**(a)**           **(b)**

**Figure 5:** Two possible latent structures that confound each pair of variables $x_i$, $x_j$ and $x_k$.

## F OPTIMISATION DETAILS

### F.1 OPTIMISATION DETAILS FOR N-ADMG

As discussed in Section 5, we gradually increase the prior hyperparameters $\rho$ and $\alpha$ throughout training. This is done using the augmented Lagrangian procedure for optimisation (). The optimisation process interleaves two steps: 1) optimise the objective for fixed values of $\rho$ and $\alpha$ for a certain number of steps; and ii) update the values of $\rho$ and $\alpha$. Steps i) and ii) and ran until convergence, or the maximum number of optimisation steps is reached. We describe these two steps in more detail below.

**Step i).** The objective is optimised for some fixed vales of $\rho$ and $\alpha$ using Adam (). We use a learning rate of $10^{-3}$ for the model parameters and $5 \times 10^{-3}$ for the variational parameters. We optimise the objective for a maxmimum of 5000 steps or until convergence (we stop early if the loss does not improve for 1500 optimisation steps, moving to step ii)). During training, we reduce the learning rate by a factor of 10 if the training loss does not improve for 1000 steps a maximum of two times. If we reach the condition a third time, we assume optimisation has converged and move to step ii). We apply annealing to the KL-divergence between the approximate posterior and prior over the latent variables. The annealing contant is fixed for each step i), and increased linearly over the first optimisation loops.

**Step ii).** We initialise $\rho = 1$ and $\alpha = 0$. At the beginning of step i) we measure the DAG / bow-free penalty $P_1 = \mathbb{E}_{q_\phi(G)}[h(G)]$. At the beginning of step ii), we measure this penalty again, $P_2 = \mathbb{E}_{q_\phi(G)}[h(G)]$. If $P_2 < P_1$, we leave $\rho$ unchanged and update $\alpha \leftarrow \alpha + \rho P_2$. Otherwise, if $P_2 \geq 0.65 P_1$, we leave $\alpha$ unchanged and update $\rho \leftarrow 10\rho$. We repeat the steps i) to ii) a maximum of 30 times or until convergence (measured as $\alpha$ or $\rho$ reaching some max value which we set to $10^3$ for both).

### F.2 ADDITIONAL HYPERPARAMETERS

**Prior Hyperparameters.** We use the sparsity inducing prior hyperparameters $\lambda_{s,1} = \lambda_{s,2} = 5$.

**ELBO approximation.** We construct an approximation to the ELBO in Equation 15 using a single sample from the approximate posteriors. For evaluating the gradients of the ELBO we use the Gumbel softmax method with a hard forward pass and soft backward pass with temperature of 0.25.

**Neural network architectures.** The functions $\xi_1$, $\xi_2$ and $\ell$ used in the likelihood and the inference network used to parameterise $q_\phi(\mathbf{u}|\mathbf{x})$ are all two hidden layer MLPs with 80 hidden units per hidden layer.

**Non-Gaussian noise model.** For the non-Gaussian noise model

**ATE estimation.** For ATE estimation we compute expectations by drawing 1000 graphs from the learnt posterior, and for each graph we draw two samples of $\mathbf{x}_Y$ for a total of 2000 samples which we used to form a Monte Carlo estimate.

## G    DATASET DETAILS

### G.1    SYNTHETIC FORK-COLLIDER DATASET

We constructed a 2000 sample synthetic dataset with the causal structure shown in Figure 1 by sampling from the following SEM:

$$[u_1,\ u_2,\ \epsilon_1,\ \epsilon_2,\ \epsilon_3,\ \epsilon_4,\ \epsilon_5]^T \sim \mathcal{N}\left(\mathbf{0}, \mathbf{I}\right)$$
$$x_1 = \epsilon_1$$
$$x_2 = \sqrt{6}\exp(-u_1^2) + 0.1\epsilon_2$$
$$x_3 = \sqrt{6}\exp(-u_1^2) + \sqrt{6}\exp(-u_2^2) + 0.2\epsilon_3 \tag{37}$$
$$x_4 = \sqrt{6}\exp(-u_2^2) + \sqrt{6}\exp(-x_1^2) + 0.1\epsilon_4$$
$$x_5 = \sqrt{6}\exp(-x_1^2) + 0.1\epsilon_5.$$

Variables $u_1$ and $u_2$ are latent confounders acting on variable pairs $x_2$ and $x_3$, and $x_3$ and $x_4$, respectively.

### G.2    LATENT CONFOUNDED ER DATASET

We generate synthetic datasets from ADMG extension of *Erdős-Rényi* (ER) graph model (Lachapelle et al., 2019; Zheng et al., 2020). An $\mathbf{ER}(d, e, m)$ dataset is generated according the following procedures:

1. Generate a $d \times d$ directed adjacency matrix $G_D$ of a ADMG from *Erdős-Rényi* (ER) graph model, whose expected directed edges equal to $e$;

2. Simulate $d \times d$ bidirected adjacency matrix $G_B$ via random Bernoulli sampling, whose expected number equals to $m$;

3. Simulate exogenous noises $\epsilon_i$ from a zero mean Gaussian distribution with standard deviation of 0.1;

4. Simulate latent variables $\mathbf{u}$ from a zero mean Gaussian distribution with standard deviation of 0.1;

5. Simulate each observed variable as $x_i = f_i(\mathbf{x}_{\text{pa}(i;G_D)}) + g_i(\mathbf{u}_{\text{pa}(i;G_B)}) + \epsilon_i$, where $f_i, g_i$ are randomly sampled nonlinear functions of the form: $y = \mathbf{w}^T e^{-\mathbf{x}^2}$.

6. Remove $\mathbf{u}$ from the sampled dataset.

### G.3    IHDP DATASET DETAILS

This dataset contains measurements of both infants (birth weight, head circumference, etc.) and their mother (smoked cigarettes, drank alcohol, took drugs, etc) during real-life data collected in a randomised experiment. The main task is to estimate the effect of home visits by specialists on future cognitive test scores of infants. The outcomes of treatments are simulated artificially as in Hill (2011); hence the outcomes of both treatments (home visits or not) on each subject are known. Note that for each subject, our models are only exposed to only one of the treatments; the outcomes of the other potential/counterfactual outcomes are hidden from the mode, and are only used for the purpose of ATE evaluation. To make the task more challenging, additional confoundings are manually introduced by removing a subset (non-white mothers) of the treated children population. In this way we can construct the IHDP dataset of 747 individuals with 6 continuous covariates and 19 binary covariates. We use 10 replicates of different simulations based on setting B (log-linear response surfaces)

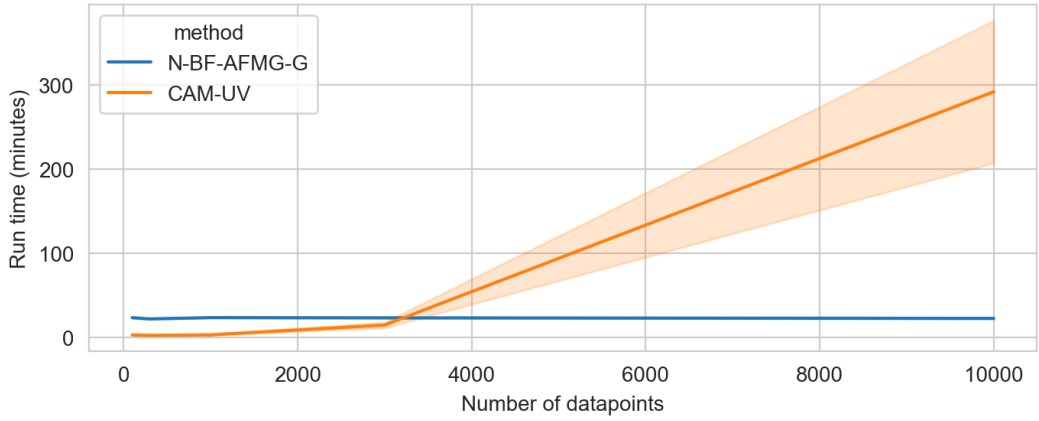

**Figure 6:** Run time results for N-BF-ADMG and CAM-UV on a 12 variable synthetic ER dataset. The figure shows mean results ± standard deviation across five randomly generated datasets.

of Hill (2011), which can downloaded from `https://github.com/AMLab-Amsterdam/CEVAE`. We use a 70%/30% train-test split ratio. Before training our models, all continuous covariates are normalised.

## H   RUN TIME COMPARISON

Here, we compare the run time of N-BF-ADMG and CAM-UV on a synthetically generated **ER**(12, 50, 10) dataset. The results are shown in Figure 6. N-BF-ADMG is trained using 30k epochs to ensure convergence, whose run time can be further reduced via early stopping etc. The results in Figure 6 highlight that methods based on continuous optimisation (N-BF-ADMG) offer a significant improvement in run time relative to methods based on conditional independency tests (CAM-UV) for large datasets. In combination with the results in the main paper, this shows the empirical validity of deep learning approach for identifying ADMGs that scales to larger dataset, without sacrificing accuracy.

