# OpenReview forum: "Causal Reasoning in the Presence of Latent Confounders via Neural ADMG Learning"
_ICLR.cc/2023/Conference — ICLR 2023 poster_

### Official Review · Reviewer_dczA · 2022-10-22

**Confidence:** 4
**Correctness:** 3
**Technical Novelty And Significance:** 2
**Empirical Novelty And Significance:** 2
**Recommendation:** 6

**Clarity, Quality, Novelty And Reproducibility:**

Overall, the quality is good. The authors take a further step towards the differentiable causal discovery when the data generating process is non-linear.


typo or minor aspects
1. They enable us to answer questions causal in nature on Page 1.
2. Lemma 1-Lemma 3 "if *and* only if "on Page 4.


questions:
1. I am not clear how the proposed method enforces that the ADMG learning satisfies
assumptions in Section 4, as said on Page 5. Shouldn't whether the assumptions are fulfilled related to the task only and not relevant to the method?

2. What does it mean by "Whilst the observed data distribution may still be identifiable"?

**Details Of Ethics Concerns:**

-

**Strength And Weaknesses:**

Strength:
1. The paper is clearly written.
2. The authors give a detailed review of the related studies.
3. There are some new theoretical results in this paper.

Weaknesses:
I do not find some significant weaknesses except for some typos.

**Summary Of The Paper:**

In this paper, the authors generalize the differentiable causal discovery method to the setting in the presence of latent variables. Compared to related work, the method can tackle non-linear data. They provide both theoretical and empirical results.

**Summary Of The Review:**

Overall, the quality is good. The authors propose a differentiable method for learning ADMG under the current causal discovery framework, with theoretical guarantees and rational experiments. Hence, I give a positive score. The main reason for the borderline score is that I do not quite agree with the study regarding learning the ADMG (this discussion is possibly beyond the scope of this paper.). The assumptions necessary for ADMG learning are too strong. I am strongly skeptical whether such a target of learning ADMG can be achieved in practice, and whether this study can help the practical problems.

---

> ### Author Response · Authors · 2022-11-14
> **Thank you for your review.**
>
> Thank you very much for your review. We are pleased to see that you appreciate the effort and novelty of our work and did not find significant weaknesses. Below we address your question respectively.
>
> > typo or minor aspects
>
> Thanks for the suggestions, we have revised our paper accordingly.
>
> > Shouldn't whether the assumptions are fulfilled related to the task only and not relevant to the method?
>
> Your understanding is correct. Whether assumptions are fulfilled is related to the task only. We have revised the paper to further clarify our relevant statements.
>
> Our method cannot enforce assumptions to hold in the real-world application itself. What we presented in Section 5 is to answer the question “how can we optimise our model in the space of ADMG graphs assuming that our data is generated by an ADMG”. This leads to the bow-free ADMG constraint in Section 5.3.
>
>
>
>
> > Assumptions necessary for ADMG learning are too strong. …whether this study can help the practical problems.
>
> We understand your concern and it is a general consideration of the entire field. We argue that we have advanced the field to make the assumptions weaker and proposed methods that are more practical in real-world applications than existing ones. More specifically, there are a few layers to your concern, and we would like to address them separately as below.
>
> The assumptions for ADMGs identification (proposition 1). Our main assumptions are: Assumption 1 (the functional form of SCMs follows Eq (5)); and Assumption 2 (Latent variables confound pairs of non-adjacent observed variables). Assumption 1, by far, is the most general form for SCM-based ADMG learning; and Assumption 2 is already general and flexible enough, in the sense that it can produce the same conditional independencies amongst the observed variables in any given causal graph (see discussions in Section 4).  Therefore, our assumptions are in fact less strong relative to many prior works.
> Usefulness in practice. Firstly, our theory contribution allowed us to propose a novel scalable and flexible framework that is closer to real-world application needs compared to prior work. Secondly, the question regarding practical usage is best answered by empirical experiments. In our synthetic and real-world data experiments we showed that it generally makes less mistakes and outperformed other ADMG/DAG baselines. This demonstrates the practical usefulness of our proposed method.
>
>
> In the end, we also extended our discussion in the last section and pointed out that further relaxing the assumptions is a key future direction.
>
> > What does it mean by "Whilst the observed data distribution may still be identifiable"?
>
> By the ‘observed data distribution’, we refer to the probability distribution over the observed variables when no interventions are made to the data generating process. To estimate causal effects, we require the distribution over observed variables after interventions have been made on the data generating process. The latter is not necessarily identifiable given the former, particularly in the presence of latent confounding.

---

> ### Author Response · Authors · 2022-11-17
> **Rebuttal discussion**
>
> Dear Reviewer dczA,
>
> We would like to thank you again for spending time carefully evaluating our submission and providing valuable feedback. We would appreciate if you could let us know if our rebuttal and revised paper addressed your concerns; and if so if you would reconsider the rating. As the initial rebuttal period is ending soon, we won't be able to revise the paper anymore. We are pleased to engage in further discussions during the following discussion period.

---

### Official Review · Reviewer_ZhRx · 2022-10-22

**Confidence:** 3
**Correctness:** 3
**Technical Novelty And Significance:** 2
**Empirical Novelty And Significance:** 2
**Recommendation:** 5

**Clarity, Quality, Novelty And Reproducibility:**

The clarity and quality of the paper is okay.
It is hard for the reviewer to evaluate the reproducibility of the paper. Because of the page limit, many model implementation details are not disclosed in the paper.

**Strength And Weaknesses:**

Strengths:
1 I like the mathematical formulation discussed in Sec. 4. The introduction of ADMG identifiability seems to be quite clear. Though the additive noise assumption and confounding assumption are simplified, the authors achieve a good balance between practicality in real-world application and technicality in mathematical reasoning. The discussions from  lemma 1-3 also make sense to me and fairly easy to understand.

2. The autoregressive approach that is implemented with MLP is well-motivated. The authors have justified the method on both  a toy and real datasets.

Weakness:
1 I think the major issue is that the authors altered the layout of the ICLR template and significantly decrease the space between sections.
I am not sure if this is fair for other authors who follow the submission guidelines.

**Summary Of The Paper:**

In this paper, the authors modeled the data structure with acyclic directed mixed graphs (ADMGs). Under the condition where latent confounding is present, a new autoregressive  flow-based approach is proposed to learn the causal and functional relations of the given data. In the experiments. the authors validate competitive performance of their approach in comparison to the baselines.

**Summary Of The Review:**

Considering the violation of ICLR template's layout and the length of the paper, I would kindly suggest submitting this paper to a journal instead.

---

> ### Author Response · Authors · 2022-11-14
> **Thanks for your review.**
>
> Thank you very much for your positive opinions for our work. We are pleased to see that you appreciate our efforts in balancing technical soundness, practical usefulness, as well as paper clarity.
>
> > I think the major issue is that the authors altered the layout of the ICLR template and significantly decrease the space between sections
>
> Thank you for pointing out this issue. We did not intend to change the ICLR template, and this accidentally happens due to a conflict in our latex preambles. We have already fixed this conflict in our revised paper. In this revised version, we are still able to keep almost all materials of our initial version with minimal edits for the rebuttal purpose. Thus, we believe this did not give us an unfair competitive advantage and thank you for pointing this out and allowing us to fix this conflict.
>
> We hope that this resolves your concern. Please let us know if there are any other questions.

---

> ### Author Response · Authors · 2022-11-17
> **Rebuttal discussion**
>
> Dear Reviewer ZhRx,
>
> We would like to thank you again for spending time carefully evaluating our submission and providing valuable feedback. We would appreciate if you could let us know if our rebuttal and revised paper addressed your concerns; and if so if you would reconsider the rating. As the initial rebuttal period is ending soon, we won't be able to revise the paper anymore. We are pleased to engage in further discussions during the following discussion period.

---

### Official Review · Reviewer_DadL · 2022-11-01

**Confidence:** 3
**Correctness:** 4
**Technical Novelty And Significance:** 4
**Empirical Novelty And Significance:** 4
**Recommendation:** 8

**Clarity, Quality, Novelty And Reproducibility:**

This paper is well written, and a rigorous discussion of existing work and its limitations is provided. Additionally, the explanation of concepts flows well and allows for the reader to understand the points made in this paper very well. The work here is novel as it provides new identification results as well as a new numerical framework for causal discovery.

**Strength And Weaknesses:**

Strengths: This paper establishes causal discovery results on a more flexible class of ADMGs than what exists in the literature. Proper identification conditions are established, and the inference framework has computational tractability advantages.

Weakness: The statistical framework for neural ADMG learning does require a lot of modelling decisions when it comes to things such as choice of variational approximation - and I do worry how how much flexibility these methods lose when such choices are made. However, I understand the necessity of these assumptions, and I consider this only a minor weakness.

**Summary Of The Paper:**

This paper provides a gradient based approach to causal discovery in the presence of unobserved confounding. By making assumptions on the structure of the graph along with additive noise assumptions - identification results for non-linear SCMs are provides. Since the conditional independencies involves learning functions, a neural ADMG learning framework is proposed. Finally, empirical evaluation is performed where the method is shown to perform well.

**Summary Of The Review:**

Overall, this paper presents a novel causal discovery algorithm under the assumption of additive noise. Identification results are first established, then the proposed neural ADMG learning framework is used to learn an ADMG. This paper is novel because it allows for learning ADMGs while relaxing assumptions made by previous works since, it allows for non-linear functions in the SCM. Next, instead of relying on traditional conditional independence tests - which can sometimes run into tractability issues in high dimensions, the neural ADMG framework is proposed that is more suitable for dealing with high dimensional data. High quality paper, I recommend for acceptance.

---

> ### Author Response · Authors · 2022-11-14
> **Thank you for your encouraging review.**
>
> Thank you for your encouraging review and valuable suggestions to improve.  We are particularly pleased that you appreciate the novelty of our work and the clarity of the presentation.
>
> Regarding the modelling choices, especially the variational approximation, we do agree that this is a very important aspect to look at. Meanwhile, this could be a deeper topic on its own. We are indeed already doing in-depth research for this topic (improving approximate inference  & uncertainty estimations for deep causal models). We have revised our future work section to extend these discussions.

---

> ### Author Response · Authors · 2022-11-17
> **Rebuttal discussion**
>
> Dear Reviewer DadL,
>
> We would like to thank you again for spending time carefully evaluating our submission and providing valuable feedback. We would appreciate if you could let us know if our rebuttal and revised paper addressed your concerns. As the initial rebuttal period is ending soon, we won't be able to revise the paper anymore. We are pleased to engage in further discussions during the following discussion period.

---

### Official Review · Reviewer_fy3p · 2022-11-03

**Confidence:** 4
**Clarity, Quality, Novelty And Reproducibility:** The paper is well-written and easy to…
**Correctness:** 4
**Technical Novelty And Significance:** 3
**Empirical Novelty And Significance:** 2
**Recommendation:** 5

**Strength And Weaknesses:**

STRENGTHS

While this paper builds heavily on existing work, both in deriving the sufficient conditions for ADMG identifiability and for specifying the variational model optimized with neural nets, the two parts fit together nicely which leads to a well-defined model for ADMG estimation.

The research area is also very relevant and has received a lot of attention recently (deep causal models, accounting for unobserved confounding).


WEAKNESSES

The novelty of the proposed approach is somewhat limited by the fact that the vital building blocks (ADMG identification criteria and variational causal graph structure learning) are already present in the literature.

The experiments appear inconclusive, especially given the fact that both in the causal discovery and causal inference (ATE estimation) experiment, the methods coming close to (or beating) N-ADMG are the ones corresponding to the "building blocks", i.e. N-DAG and CAM-UV.

As a minor issue, Figures 2 and 3 are difficult to read.


**Summary Of The Paper:**

This paper introduces a method of estimating ADMGs from observational data. The authors build on existing work on ADMG identifiability and on graph learning with deep neural networks. First, two assumptions (sufficient conditions) are proposed that guarantee identifiability of the underlying ADMG (inspired by Maeda and Shimizu, 2021). Subsequently, a variational scheme (N-ADMG) for learning the ADMG is introduced (based on Geffner et al., 2022). The paper concludes with a series of experiments showing causal discovery (ADMG recovery) and causal inference (average treatment effect estimation).

**Summary Of The Review:**

This paper combines two existing ideas which leads to a somewhat novel approach to ADMG estimation from observable data.

---

> ### Author Response · Authors · 2022-11-15
> **Thank you for your constructive review.**
>
> Thank you for the constructive feedback and detailed suggestions.Please see the general reply above where we listed all changes that we made and clarified our contribution.
> Below, we respond to each of your comments.
>
> > The novelty of the proposed approach…building blocks (ADMG identification criteria and variational causal graph structure learning) are already present in the literature.
>
> Our new theory is indeed an extension and build on existing work. However, we would like to emphasise the importance of this theory to our key contribution of a novel gradient-based approach to learning a flexible, nonlinear ADMG  model.  Our work considers the real-world motivation of modelling causal relationships in the presence of latent confounding where the flexible and nonlinear models are needed for large-scale applications.
>
> Our theoretical contribution extends the existing ADMG identification condition from the literature to more general non-linear SCMs. To the best of our knowledge, this is a new result that applies for the most general SCM form in the literature, which shows the theoretical validity of developing a flexible and scalable deep learning based approach for ADMG identification. Thus, this theoretical result is significant for method development for the community although it follows similar proving techniques.
>
>
>
> Besides the theoretical contribution, we contributed a novel algorithm.  Our practical algorithm is fundamentally different from existing ADMG-based methods e.g. RCD and CAM-UV, where they are based on regression+independence test whereas ours rely heavily on deep generative models and differentiable optimization. Note also that existing ADMG methods like RCD and CAM-UV *cannot* directly perform causal inference. In Experiment 6.3, the ATE estimations from these baselines are obtained by applying *our method* to learning a non-linear flow-based ANM ($q(G)$ is fixed to deterministic ADMG discovered by RCD/CAM-UV) to estimate ATEs.
>
> > The experiments appear inconclusive… the methods coming close to (or beating) N-ADMG are the ones corresponding to the "building blocks", i.e. N-DAG and CAM-UV.
>
> There may be misunderstanding regarding how to interpret the results. We believe that, on the contrary, the experiment results actually showed the necessity and advantage of our method.
>
> Generally, the more realistic the method’s assumptions are, the better its performance will be. For example, CAM-UV is one of our strongest baselines which consistently outperformed many existing baselines (except for ours). This is because its assumption (ADMG with fully decoupled interactions) is more general than  RCD, DCD and N-DAG (linear interactions). This is exactly why we developed our methods to further generalise the assumptions of CAM-UV (fully decoupled nonlinearities) to more general ADMG SCMs (general nonlinearities), and introduce deep generative models to address its issues in scalability. In our real-world data experiments in 6.3, it clearly shows the advantage of our method (having more flexible assumptions) as it consistently outperforms CAM-UV.
>
> In Addition, in Appendix H, we have added a run-time comparison, showing that our method is massively more scalable than CAM-UV. In combination with the results in the main paper, this is the first evidence in the literature that shows the empirical validity of deep learning approach for identifying ADMGs that scales to larger dataset, without sacrificing accuracy.
>
> > As a minor issue, Figures 2 and 3 are difficult to read.
>
> We have updated Figures 2 and 3 so that they are easier to read.
>
> We hope that the replay resolves your concern. Please let us know if you have any further questions.

---

> ### Author Response · Authors · 2022-11-17
> **Rebuttal discussion**
>
> Dear Reviewer fy3p,
>
> We would like to thank you again for spending time carefully evaluating our submission and providing valuable feedback. We would appreciate if you could let us know if our rebuttal and revised paper addressed your concerns; and if so if you would reconsider the rating. As the initial rebuttal period is ending soon, we won't be able to revise the paper anymore. We are pleased to engage in further discussions during the following discussion period.

---

### Author Response · Authors · 2022-11-15
**Rebuttal Summary**

Dear reviewers,

We would like to thank you for your time considered to review our paper and provide insightful feedback. We acknowledge that the reviewers highlighted the importance of our motivation (fy3p, DadL, ZhRx), the significance and technical soundness of our proposed method ( fy3p, DadL, ZhRx, dczA) and the extensiveness of empirical evaluations (DadL, ZhRx).

We have addressed all your concerns and answered personally to each of your reviews. We uploaded a new revised version of the paper that includes all the suggested changes. We encourage discussion on the rebuttal and revised paper, in order to address more concerns in case you still have any. We hope that after we have addressed all your concerns for improving our paper, you could reconsider your rating.

We list below the summary of changes introduced in the revision version (all changes are highlighted in blue in the revised paper):

- We have clarified our relevant statements on assumptions (Section 5) as suggested by dczA.
- We have revised our future work (Section 7) to extend these discussions, especially on variational approximations, as suggested by reviewer DadL.
- We have added a new run-time comparison in response to reviewer fy3p, showing that our method massively improved the run-time - performance over CAM-UV.
- We have revised Figures 2 and 3 accordingly as suggested by reviewer fy3p.
- We have fixed the template conflict, as suggested by reviewer ZhRx. In this revised version, we are still able to keep almost all materials of our initial version with very minor edits for the rebuttal purpose.
- We have fixed a few typos as suggested by reviewer dczA.

---

### Comment · Area_Chair_JJfa · 2022-12-07
**Response to Author Feedback**

Dear Reviewers, thank you so much again for your time on this paper. The discussion phase is still ongoing, how does the author response and other reviews change your view of the paper?

---

### Decision · Program_Chairs · 2023-01-20

**Decision:**

Accept: poster

**Justification For Why Not Higher Score:**

Even though I find it promising, it is likely only interesting to a specialized audience.

**Justification For Why Not Lower Score:**

The paper makes useful contributions in causal identification and estimation, accepting it will improve the conference.

**Metareview: Summary, Strengths And Weaknesses:**

The reviewers were split about this paper and did not come to a consensus: on one hand they appreciated the importance of the problem addressed and the clarity of the writing, on the other they questioned whether the method was sufficiently novel. After going through the paper and the discussion I have decided to vote to accept for the following reasons: (a) the identifiability result presented in the paper is non-trivial and extremely useful for the field (adding a new identifiability scenario beyond linearity or fully decoupled effects), (b) the gradient-based optimization framework is highly-practical and would be easy to implement in any autodiff framework. The authors will release their code which will be a great contribution to the community. Authors: you've already indicated that you've updated the submission to respond to reviewer changes, if you could double check their comments for any recommendation you may have missed on accident that would be great! The paper will make a great contribution to the conference!

**Note From Pc:**

if the above contains the word "oral" or "spotlight" please see: "oral" presentation means -> notable-top-5% and "spotlight" means -> notable-top-25%. As stated in our emails, we are disassociating presentation type from AC recommendations